# Morphology Observation of Two-Dimensional Monolayers of Model Proteins on Water Surface as Revealed by Dropping Method

**DOI:** 10.3390/bioengineering11040366

**Published:** 2024-04-11

**Authors:** Yukie Asada, Shinya Tanaka, Hirotaka Nagano, Hiroki Noguchi, Akihiro Yoshino, Keijiro Taga, Yasushi Yamamoto, Zameer Shervani

**Affiliations:** 1Department of Life Science and Applied Chemistry, Graduate School of Engineering, Nagoya Institute of Technology, Nagoya 466-8555, Japan; 2Food & Energy Security Research & Product Centre, Sendai 980-0871, Japan

**Keywords:** gramicidin-D (GD), alamethicin (Al), monolayer, dropping method (DM), surface tension measurement (STm), Brewster angle microscopy (BAM), atomic force microscopy (AFM)

## Abstract

We have investigated the morphology of two-dimensional monolayers of gramicidin-D (GD) and alamethicin (Al) formed on the water surface by the dropping method (DM) using surface tension measurement (STm), Brewster angle microscopy (BAM), and atomic force microscopy (AFM). Dynamic light scattering (DLS) revealed that GD in alcoholic solutions formed a dimeric helical structure. According to the CD and NMR spectroscopies, GD molecules existed in dimer form in methanol and lipid membrane environments. The STm results and BAM images revealed that the GD dimer monolayer was in a liquid expanded (LE) state, whereas the Al monolayer was in a liquid condensed (LC) state. The limiting molecular area (*A*_0_) was 6.2 ± 0.5 nm^2^ for the GD-dimer and 3.6 ± 0.5 nm^2^ for the Al molecule. The AFM images also showed that the molecular long axes of both the GD-dimer and Al were horizontal to the water surface. The stability of each monolayer was confirmed by the time dependence of the surface pressure (*π*) observed using the STm method. The DM monolayer preparation method for GD-dimer and Al peptide molecules is a useful technique for revealing how the model biological membrane’s components assemble in two dimensions on the water surface.

## 1. Introduction

Biological membrane is a general term used for membrane structures that constitute a number of cell structures, such as intracellular vesicles and plasma membranes. Membranes are mainly composed of lipid bilayers with proteins buried in them. Their functions include working as barriers, substance transport, molecular recognition, and signal transduction. The change in state between the fluid and rigid states is related to the process of various cell functions [1,2,3,4]. Membrane proteins, constituting about 20 to 80% of the membranes, play important roles in enzyme activity, transporting the components, acting as receptors, and forming the pores in cell membranes [2,5]. Among the membrane proteins, gramicidin and alamethicin, used in this research, have been studied widely as peptide models to study proteins’ amphiphilic properties and understand the interactions between lipids and protein polymorphisms that occur as they adapt to the membrane environment [2]. Gramicidin is a hydrophobic chain peptide that forms ion channels by aggregating two molecules in lipid bilayer membranes [6,7,8]. Depending on the surrounding environment, 15 amino acids alternately arranged between L- and D-type amino acids, forming a double-helix structure also known as the β-helix. On the other hand, alamethicin is an amphiphilic chain peptide that acts as an ion channel-forming protein by assembling 3–12 molecules within the lipid bilayer membrane [9,10]. It is composed of 20 amino acids and forms a bent α-helix in a hydrophobic environment.

A number of studies have been carried out to understand the function of biological membranes using model membranes such as monolayer, bilayer, micelle, vesicle, and lamellar structures [1,2,3,11,12]. The monolayer on the water surface has several advantages over other models. Firstly, it is easy to prepare, and two-dimensional investigations can be conducted. Secondly, the number of monolayer-forming molecules added to the water surface can be easily controlled. Lastly, the monolayer has a structure that is half that of the real biological membrane, thereby providing advantages for studying parameters that cannot be measured in a complete membrane. Two different methods are used to prepare monolayers on water surfaces [13]. The first is the compression method (CM), in which monolayer molecules are studded on the water surface and then compressed with a partition board to form the monolayer. It is possible to form a high-density monolayer by promoting the original intermolecular interactions among the monolayer molecules through compression using an artificial power supply. The second is the dropping method (DM), in which monolayer molecules are studded continuously on the water surface to form the monolayer. A spontaneous monolayer is formed by maintaining sufficient interactions between the hydrophilic groups of the monolayer and the water molecules. It is possible to form a highly elastic and fluid monolayer using the DM.

We investigated the morphology of various phospholipid monolayers using the DM. The dipalmitoyl phosphatidyl choline (DPPC) monolayer showed a semi-expanded state and had a fluidic structure that was different from that of a condensed monolayer obtained using the CM [13,14]. Such a fluidic DPPC monolayer also showed a specific condensation behavior through the action of cholesterol [13,15]. The addition of cholesterol in the dimyristoyl PC (DMPC) monolayer caused the transition from a liquid expanded (LE) state to a liquid condensed (LC) state with the structural transition (gauche-to-trans conformation) of alkyl chains in DMPC [16] molecules. It was also reported that DPPC and DMPC mixed monolayers had the properties of pure DPPC at low mole fractions of *x_DMPC_* and neat DMPC at high x_DMPC_, which showed their three-dimensional bulk structural properties [17]. In the above reports [13,14,15,16,17], we discussed the applications of the monolayers prepared by the DM method as a model biological membrane. Biological membrane constituents, such as phospholipids and membrane proteins, express and perform their functions using the surrounding water as a medium. The study of the air–water interface mediated by water molecules elucidates the functioning of the membranes and provides information about their properties and specificities. In this research, protein monolayers of gramicidin (GD) and alamethicin (Al) were prepared using a dropping method similar to that used for phospholipid monolayers. We used surface tension measurement (STm), Brewster angle microscopy (BAM), and atomic force microscopy (AFM) to examine the physicochemical properties of the monolayers.

## 2. Materials and Methods

### 2.1. Materials

Gramicidin-D (GD), a mixture of gramicidin-A, -B, and -C (GA: 80%, GB: 6%, GC: 14%, purity > 95%, source *Bacillus brevis*), was purchased from Sigma-Aldrich Co. LLC (St. Louis, MO, USA). Alamethicin (Al, purity 99%) was from LKT Laboratories Inc. (St Paul, MN, USA). Figure 1 shows the structures of GD and Al. These were used without further purification. The spreading solvent used for monolayer preparation was methanol for GD and a methanol and chloroform (1:2; *v*:*v*) mixture for Al. The purity of methanol and chloroform was 99.5% and 99%, respectively. Both methanol and chloroform were obtained from Wako Pure Chemical Ind. Ltd., Osaka, Japan. The water used to prepare the monolayer had a conductance of <0.07 μS/cm. It was purified using a Super Water Purifying System (WL-21P; Yamato Scientific Corp. Ltd., Tokyo, Japan).

### 2.2. Methods

#### 2.2.1. Monolayer Formation

GD and Al were dissolved in the corresponding solvent to prepare dropping solutions of 0.1 mM. Each solution was spread on a purified water surface using a 50 μL syringe (Hamilton Corp., Hamilton, NY, USA) to prepare the monolayer. The details of monolayer preparation using the DM have been reported in our previous articles [13,14,15,16,17]. In brief, a drop of 1 μL volume solution was gently put on the surface of the water. The next drop was added after a time lag ≥ 1 min when the previous drop achieved expanding equilibrium on the water surface. After the droplet expanded and surface tension became constant (mentioned in Section 3.1.), the value was recorded. The above process was repeated until the lens on the water was formed and the surface tension value remained unchanged even after additional drops were added. The completion of monolayer formation was confirmed when the surface tension value became constant.

#### 2.2.2. Surface Tension Measurement (STm)

Surface tension measurement (STm) for the GD and Al monolayers was carried out using a Surface Tensiometer (CBVP-A3; Kyowa Interface Science Corp. Ltd., Niiza, Japan) fitted with a Wilhelmy plate (platinum) [13,14]. Surface tension was recorded after spreading each droplet at a constant temperature of 26 ± 1 °C. The surface tension value was converted to surface pressure (*π*) using the following equation:*π* = *γ*_0_ − *γ*
where *γ*_0_ is the surface tension value of the water surface and *γ* is the value after spreading the droplet. The *π*-*A* curves (*A*: molecular area) of the monolayers were constructed as a function of the dropping volume (molecule numbers).

#### 2.2.3. Brewster Angle Microscopy (BAM)

Brewster angle microscopy (BAM) for visualizing and capturing the morphology of the GD and Al monolayers was carried out using a BAM microscope (EMM633K; Filgen Inc., Nagoya, Japan) equipped with USB-CAP-type (GV-USB2; I-O DATA DEVICE, Inc., Ishikawa, Japan) image analysis software [13,14,15,16,17,18]. BAM showed the monolayer image that was formed due to the difference in refractive indices between the monolayer and the pure water phase. A BAM microscope was fixed on a glass dish, and *p*-polarized light of wavelength 632.8 nm from a He-Ne laser of 10 mW was irradiated at a Brewster angle of 53.1° on the surface of the monolayer and neat water. A 40 mm focal-length lens reflected light was magnified and detected by a CCD camera (Hamamatsu Photonics C5948-70, Hamamatsu, Japan). At 26 ± 1 °C, both monolayers prepared by the above methods (Section 2.2.1) were observed in real time. The lateral resolution of the image was about 1 μm.

#### 2.2.4. Atomic Force Microscopy (AFM)

Atomic force microscopy (AFM) was used to investigate the morphology of the GD and Al monolayers. A commercial 1.5 kV-AFM microscope (JSPM-5200; JEOL Ltd., Tokyo, Japan) and JEOL-original SPS mapping software (vol.54) in contact mode using a 10 μm tube-type scanner were employed. Both vertical and lateral deflections were detected by an optical beam deflection method with a four-segment photodetector to observe topographic and frictional forces, respectively. The resolution of the AFM image was about a mica atomic image. An HOPG (highly oriented pyloric graphite) substrate of area 20 × 20 mm^2^ was attached to the tip of the auto-elevation system and dipped into the water phase in a glass dish. The GD and Al monolayers were formed on the water surface, and after more than 45 min following the completion of monolayer formation, the HOPG substrate in water was horizontally scooped from below the monolayer using the scooping-up method [13,14,15,19]. With a 1 mm/min transfer velocity, the monolayer was transferred to the substrate at 26 ± 1 °C.

## 3. Results and Discussion

### 3.1. π-A Isotherm Curve

Figure 2a,b show the *π* (surface pressure) versus molecular area (*A*) *π-A* isotherm curves of the GD and Al monolayers, respectively, on the water surface at 26 °C. The isotherms were constructed using the DM method. The horizontal axis represents *A* calculated from molecular numbers in the dropping volume, and the vertical axis is *π* recorded after dropping the sample solution on the water surface. The value of *π* became constant after the monolayer reached equilibrium, and the reading of *π* was noted. For comparison, we also show the curve of the GD and AI monolayers obtained using the CM (- - -) (in both the figures a and b). A computerized Langmuir–Blodgett (LB) Trough (USI-3-77, USI Corp., Fukuoka, Japan) was used for the formation of each monolayer by adjusting the trough area to 556 × 149 mm^2^ and the barrier speed to 5 cm^2^/min.

In methanol solution, GD molecules form double-helix structures (dimers), which was confirmed by recording CD spectra using a spectrometer J-820 (JASCO Corp., Tokyo, Japan), a negative peak of 214 nm and a positive peak of less than 207 nm value of ellipticity (*θ*) were reported in an experiment conducted in our laboratory. Similar results were reported by Chen et al. [20] and Chaudhuri et al. [21]. Michielsent and Pecora [22] also reported a dimeric helical structure for GD in alcoholic solutions. According to nuclear magnetic resonance (NMR) and circular dichroism (CD) spectroscopies [23], GD molecules form helical dimer structures in methanol and lipid membrane environments. In article [24], the CD spectrum of GD dimers was reported. In a lipid environment, two monomers form a dimer by joining at their N-termini. Taking into account the dimeric structure of GD and the *π-A* isotherm curve using the DM (o) and the CM (- - -), as shown in Figure 2a, the dimer mass area was considered (horizontal axis) to construct the isotherm. Plot (○) in Figure 2a shows that *π* increased gradually from a molecular area > 8.0 nm^2^ and kept increasing monotonously up to 16 mN/m. The shape of the curve was smooth, and the limiting molecular area (*A*_0_) was 6.2 ± 0.5 nm^2^/dimer (14 mN/m). The recorded *A*_0_ and *π* values suggest that the GD monolayer is in the LE state. Plot (- - -) in Figure 2a was different from plot (○). *π* increased gradually from a molecular area > 9.0 nm^2^. After a plateau region at 20 mN/m, *π* increased up to 40 mN/m. *A*_0_ was 3.8 nm^2^/molecule (38 mN/m). Figure 2a shows similarities to the findings of Volinsky et al. [25].

It is known that Al molecules form a single helix with a rigid structure that bends slightly near the middle of the molecule (the 14th proline residue) [2,26]. Alamethicin is a peptide composed of 20 amino acid residues. It contains 7-α-aminoisobutyric acid residues, two glutamine residues, and one free carboxyl group [27,28]. Plot (☐) in Figure 2b shows a surface pressure of almost 0 mN/m until reaching 4.0 nm^2^ of molecular area, and then rises steeply up to 27 mN/m. *A*_0_ was 3.6 ± 0.5 nm^2^/molecule at 23 mN/m. The shape of the curve is quite similar to that of the cholesterol (Chol) monolayer [13,15]. The recorded *A*_0_ and *π* values indicate that the Al monolayer was in the LC state. Plots (- - -) and (☐) in Figure 2b are similar; *π* was almost 0 mN/m until reaching 3.8 nm^2^ of molecular area, and then rose steeply up to 32 mN/m. After that, *π* increased slightly and gradually to approach 35 mN/m, and *A*_0_ was 3.5 nm^2^/molecule at 24 mN/m. Plot (- - -) in Figure 2b has similarities to the findings obtained by Volinsky et al. [25,29].

### 3.2. Compression Moduli (C_s_^−1^)

Conventionally, compression moduli (*C_s_*^−1^) are used to know the physicochemical state of a Langmuir-type monolayer [30,31,32,33]. As described in the above studies, the monolayer state can be categorized into four states: solid (S), gaseous (G), liquid condensed (LC), and liquid expanded (LE). The values of *C_s_*^−1^ were <12, 12–100, 100–250, and >250 mN/m, corresponding to the G, LE, LC, and S states, respectively. From the *π*-*A* isotherms of GD and Al (Figure 2a,b), we calculated *C_s_*^−1^ using the following formula:*C_s_*^−1^ = −*A(*d*π/*d*A)*
where *A* represents the molecular area (nm^2^/molecule) and *π* is the surface pressure (mN/m). From the respective π-A isotherm curves (Figure 2a,b), the *C_s_*^−1^ versus π profiles of the GD and Al monolayers were constructed as shown in Figure 3a,b, respectively. The horizontal axis represents *π*, and the vertical axis represents the calculated *C_s_*^−1^ values. The mass area of dimer GD was taken into account when calculating the *C_s_*^−1^ value of GD (Figure 3a).

Plot (○), constructed using the DM (Figure 3a), shows that there was only one process. The *C_s_*^−1^ value increased gradually and monotonously up to *π* = 10 mN/m and subsequently decreased gradually until reaching *π* = 15 mN/m, when the monolayer formation was completed. The maximum *C_s_*^−1^ value was 25 mN/m and referred to the LE state. The profile was characteristic of the LE monolayer. The GD monolayer prepared using the DM was of the LE type, as mentioned in Section 3.1. Plot (•••), constructed using the CM (Figure 3a), was different from plot (○) and also had a double-convex-type curve. In the beginning, the *C_s_*^−1^ value increased gradually and monotonously up tol *π* = 12 mN/m and subsequently started decreasing until reaching *π*= 23 mN/m (*C_s_*^−1^ = 7 mN/m). The maximum *C_s_*^−1^ value at the first convex curve was 38 mN/m, which was due to the LE state. Next, the value increased again gradually and monotonously up to *π* = 36 mN/m and subsequently decreased until reaching *π* = 41 mN/m, when the monolayer formation was completed. The maximum value of *C_s_^−^*^1^ at the second convex curve was 40 mN/m, which corresponds to the LE state. This indicates that even the GD monolayers prepared using the CM were in the LE phase despite the two-step isotherm curves, including the plateaus. The result was quite different from that of a typical double-convex curve, such as a DPPC monolayer [31,32,33], where the first convex curve (lower *π* region) appears in the LE-type phase and the second (middle-to-high *π*) in the LC-type phase.

Plot (☐), constructed using the DM (Figure 3b), shows that there was only one process and was largely different from plot (○) of GD in Figure 2a. The *C_s_*^−1^ value increased monotonously and steeply until 24 mN/m, then decreased to more than 24 mN/m upon the completion of monolayer formation. The maximum *C_s_*^−1^ value was about 250 mN/m, which corresponds to the LC state. The profile was characteristic of the LC monolayer, indicating that the Al monolayer obtained using the DM is certainly of the LC type, as mentioned in Section 3.1. Plot (•••), constructed using the CM (shown in Figure 3b), demonstrates that there was only one process, similar to plot (☐), in which *C_s_*^−1^ increased monotonously and steeply until reaching 18 mN/m, after which the surface pressure decreased and the monolayer formation was completed. The shape of the curve was similar to that of an upward convex quadratic curve. The maximum *C_s_*^−1^ value was 250 mN/m and was recorded in the LC state. Thus, the Al monolayer exhibited the LC type regardless of the monolayer formation method. This result was similar to that of the typical one-process curve observed in cholesterol monolayers [34].

### 3.3. BAM (Brewster Angle Microscopy)

Figure 4 shows a series of BAM images of the GD (Figure 4a–c) and the Al (Figure 4d–f) monolayers on the water surface recorded at 26 °C. BAM recording was carried out for each monolayer after the dropping volume was adjusted to the three *π* values shown in the inset of the figures.

In Figure 4a–c of the GD monolayer, Figure 4a (*π* = 1 mN/m) shows a homogeneous and slightly bright contrast compared to that of the pure water surface. With increasing *π* values, the image became gradually brighter. Figure 4b (*π* = 7 mN/m) is a little brighter than Figure 4a (*π* = 1 mN/m), demonstrating that the GD monolayer continued to spread homogeneously on the water surface. Figure 4c (π = 14 mN/m) is even brighter with homogeneous contrast, revealing that homogenous GD monolayer formation was completed on the water surface. The pattern of Figure 4a–c indicate that the GD monolayer formed by the DM was in an expandable state. The result also corresponds to that shown in plot (○) of Figure 2a.

As shown in Figure 4d–f of the Al monolayer, the images are different from those in Figure 4a–c of the GD monolayer. Figure 4d of the monolayer composition (π = 0 mN/m and 9.0 nm^2^ of molecular area) was in dark contrast and nearly as dark as that of the neat water surface. In contrast, Figure 4e (*π* = 1 mN/m) has several mono- and cross-type rod-like domains, indicating strong intermolecular forces between the Al molecules. In Figure 4f (23 mN/m), a homogeneous bright picture can be observed, indicating that the formation of the Al monolayers was completed on the water surface. The patterns in Figure 4d–f are similar to those reported by Volinsky et al. using the CM preparation [25,29], with only Figure 4f being a little different. The resolution of our BAM apparatus might be somewhat low, so it could not record a clear image such as those reported in [25,29] depicting stone pavement patterns. From our above observations, shown in Figure 4d–f, it is evident that the Al monolayer formed by the DM had a condensed phase. The result was similar to that shown in plot (☐) (Figure 2b).

### 3.4. AFM (Atomic Force Microscopy)

Figure 5 shows a series of AFM images of the GD (Figure 5a,b) and the Al (Figure 5c) monolayers scooped on the HOPG surface after the completion of the monolayer using the DM at 26 °C. The AFM observation was performed on each scooped monolayer after the dropping volume was adjusted to the *π* values of two (GD) and one (Al) monolayers, as shown in the inset of each figure. The observation range was 2 μm × 2 μm. The arrows in the image and the cross-section profile below the image show the thickness (height) of each monolayer within the arrow range.

Figure 5a (*π* = 1 mN/m, GD monolayer) shows wave-type structures with a series of islands and partial defects. The contours of the AFM are shown at the bottom of the images. From the contour of Figure 5a, the height of the structures was measured to be 1.3 nm, which corresponds to the size of the GD molecule. The image shows the initial aggregation state of the GD molecules in which the molecules began to aggregate to form the monolayer. In addition, there are straight lines (bottom left) that belong to the steps of the HOPG substrate. Figure 5b (at *π* = 12 mN/m) shows a denser island-type and reduced-defect structure compared to Figure 5a. The height of the contour is the same as in Figure 5a at 1.3 nm. The existence of defects indicates that the GD monolayer at *π* = 12 mN/m was in a somewhat fluid state corresponding to that of the monolayer, which was not well transferred at the steps of HOPG (upper left of the image).

In Figure 5c of the Al monolayer, the image at *π* = 23 mN/m shows dense island-type and somewhat defective structures. The height of the structures is 1.0 nm, which corresponds to the size of the Al molecule. Therefore, this image shows a densely packed state with almost no space. The Al molecules had various two-dimensional orientation orders on the water surface. They formed a dense aggregate with the same orders, but they were less likely to interact with different orders. The difference between those orders appeared as defects when the HOPG substrate was scooped.

### 3.5. Morphology of GD and Al Monolayers

#### 3.5.1. GD Monolayer

A GD dimer with a double-helix structure has the specific characteristic that both ends of the dimer are hydrophilic and the side is hydrophobic. This specificity leads to an unstable state of dimers on the water surface. In other words, not only a weak interaction with the water molecules occurs, but molecules also have free movement on the water surface. As for the DM, the GD dimer was added little by little to the water surface and placed in the most comfortable position to prepare the monolayer. The use of the DM and the above specificity of the GD dimer tended to spread the dimers easily with weak interaction between the dimers on the water surface, thereby leading to the formation of the LE monolayer. The monolayer formation process of the GD dimer is different from that of general amphiphilic molecules. An *A*_0_ of 6.2 nm^2^ (as measured by STm) and a height (monolayer thickness) of 1.3 nm (as determined by AFM) showed that the GD dimer is oriented horizontally on the water surface, whereas the value of *A*_0_ is larger than the original molecular area of 4.5 nm^2^ calculated from a helix diameter of 1.5 nm and a molecular (dimer) long axis of 3.0 nm [2,21]. This indicates that the GD dimers in the GD monolayer have an excluded area and maintain a proper distance from each other.

On the other hand, to prepare a monolayer using the CM, the GD dimers were compressed artificially by the partitioning board after studding with dimers on the water surface. This action on the GD dimers activated the motion of the dimers and generated the interaction between them at the initial stage of the monolayer formation, resulting in an earlier increase in *π* compared to that obtained using the DM. The morphology of the monolayer until the plateau region of 20 mN/m was of the LE state, since both shapes of the *π*-*A* isotherm curve (Figure 2a) and *C_s_*^−1^ (Figure 3a) obtained using the CM were closely similar to those obtained using the DM. At the plateau region and during the subsequent increase in *π* (the completion of the GD monolayer), a change in the orientation of the GD dimer from a horizontal to a vertical direction and a densification of the monolayer occurred on the water surface. Volinsky et al. [25,29] reported a similar interpretation in which the plateau region was a phase transition from a single-stranded GD molecule to a double-stranded GD dimer, and an orientation change in the GD dimer from a parallel to a vertical direction occurred on the water surface.

We found a time dependence in *π* after the completion of the GD monolayer using the CM. Therefore, the value of *π* = 40 mN/m upon completion of the monolayer decreased rapidly and gradually and approached *π* = 21 mN/m (plateau region) within an hour. The GD dimer in the vertical direction on the water surface was somewhat unstable because an external force was applied by the partitioning board to the monolayer and one side of the hydrophilic group in the dimer faces the air phase. Actually, the orientation angle of the GD dimer from the water surface was 32.5°, calculated from the dimer’s long axis value of 3.0 nm and an *A*_0_ of 3.8 nm^2^. Therefore, such an unstable state led to the collapse of the GD monolayer after the completion of the monolayer, and partial stacking of the GD dimer (aggregates) or reorientation in a horizontal direction occurred on the water surface. This corresponded to the decrease in *π*. The *π* value after the completion of the monolayer constructed using the DM, on the other hand, was maintained at 16 mN/m after the monolayer formation. The GD monolayer constructed using the DM was formed by the spontaneous motion of the GD dimers on the water surface, resulting in a horizontal orientation of the dimers with a proper distance between them. Such behavior led to the expandable and stable GD monolayer. This corresponded to the maintenance of *π*.

#### 3.5.2. Al Monolayer

Al molecules forming a single-helix structure have the characteristic of having many hydrophilic amino acid residues. Each hydrophobic and hydrophilic side part is separated along the molecular long axis. Al is a typical amphiphilic and rigid molecule, and this characteristic leads not only to an easy interaction and arrangement between the molecules but also to the interaction with water molecules on the water surface. As for the DM, the Al molecules were added little by little to the water surface and aggregated through the interaction not only between the hydrophilic and hydrophobic side parts of the molecules but also between the hydrophilic parts and the water molecules. Al domains of various shapes were observed by BAM and aggregated spontaneously, which led to the formation of the LC monolayer. An *A*_0_ of 3.6 nm^2^ (as measured by STm) and a height (monolayer thickness) of 1.0 nm (as determined by AFM) showed that the Al molecules oriented horizontally on the water surface, whereas the value of *A*_0_ was slightly larger than the original molecular area of 3.2 nm^2^ calculated from a helix diameter of 1.0 nm and a molecular long axis of 3.2 nm [2,26]. The domains with an aligned molecular long axis formed tightly packed structures, whereas those with various azimuthal angles of molecules formed partially and slightly spaced structures. The combination of those domains corresponded to a slightly large *A*_0_. As for CM, the Al monolayer was formed in a formation process similar to that taking place when using the DM. However, an artificial force caused the forcible interaction between the Al molecules without spontaneous interactions among them, thereby leading to the formation of aggregates containing partial defects. This is also connected to the observation results of stone pavement domains by Volinsky et al. [25,29]. The artificial force applied by the CM promoted the formation of a more condensed monolayer compared to that applied by the DM.

We also found a time dependence in *π* after the completion of the Al monolayer on the CM, similar to that of the GD monolayer. Therefore, the value of *π* = 35 mN/m at the completion of the monolayer decreased rapidly and, subsequently, gradually approached *π* = 27 mN/m within an hour. The final value was near the maximum value recorded using the DM, similar to that of the GD monolayer. The slight and gradual increasing process of *π* after the steep increase in plot (- - -) in Figure 2b was due to typical excess compression, with the Al molecules being stacked on the water surface. Such an unstable state led to more stacking of molecules due to the perturbative compression effect or the collapse of stacked aggregates after compression ended. This corresponded to the decrease in *π*. The *π* value after the completion of the monolayer using the DM, on the other hand, was maintained at 27 mN/m, as was the case with the GD monolayer. The Al monolayer obtained using the DM was formed by spontaneous interaction and proper arrangement between the Al molecules. Such behavior led to the condensed and stable Al monolayer. This corresponded to the maintenance of the *π* value.

## 4. Conclusions

In this research, we have investigated the morphology of GD and Al monolayers formed on the water surface at 26 °C by using the DM (dropping method). The *π-A* isotherm curves and *C_s_*^−1^ were constructed using STm. Both STm and BAM observations showed that the GD dimer formed an LE-type monolayer and Al formed an LC-type monolayer. AFM observation also showed that both GD dimers and Al molecules orient their molecular long axis horizontally to the water surface. Comparing the results with the compression method (CM), it was found that GD dimers had the capability of easily changing their structure and orientation according to the surrounding environment due to their molecular specificity, whereas Al molecules had a high capacity for self-assembling due to the strong interaction among them. The DM was useful in revealing two-dimensional aggregation properties such as condensation, expandability, and interaction with water molecules with not only phospholipids but also protein molecules on the water surface. In our study, we found that the DM was suitable for preparing a fluid monolayer, similar to other biological membranes.

## Figures and Tables

**Figure 1 bioengineering-11-00366-f001:**
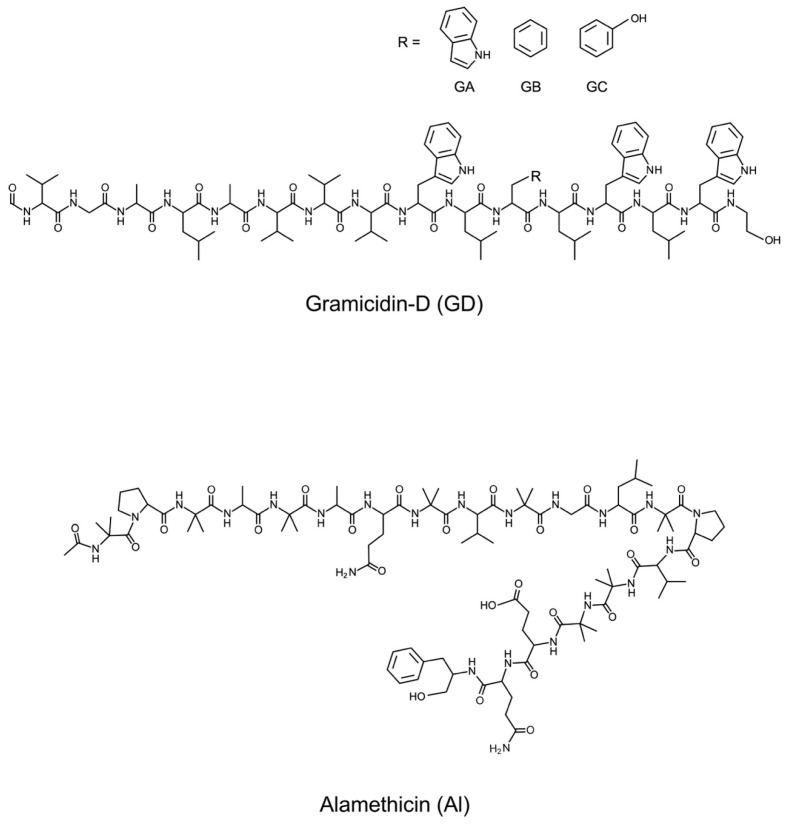
Structures of gramicidin-D (GD) and alamethicin (Al).

**Figure 2 bioengineering-11-00366-f002:**
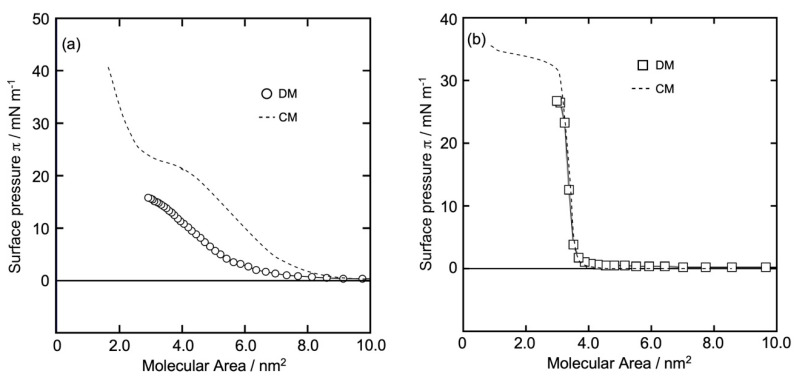
*π-A* isotherm curves of GD and Al monolayers on the water surface at 26 °C using the DM. (**a**): GD (dimer) monolayer (○); (**b**): Al monolayer (☐). (- - -) in (**a**,**b**): *π-A* isotherm curves of each GD and Al monolayer using the CM for comparison, respectively.

**Figure 3 bioengineering-11-00366-f003:**
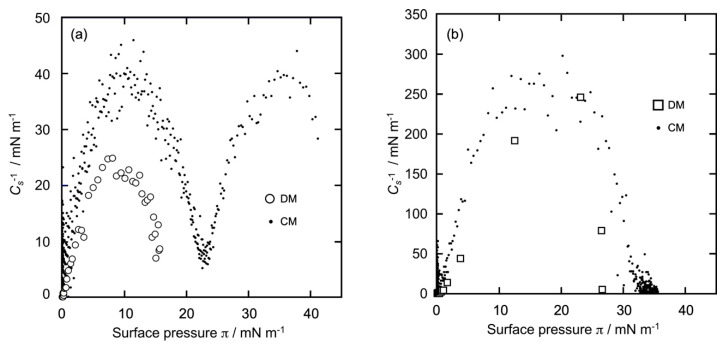
*C_s_*^−1^ profiles of GD and Al monolayers calculated from Figure 2. (**a**): GD (dimer) monolayer (○); (**b**): Al monolayer (☐). (• • •) in (**a**,**b**): *C_s_*^−1^ profiles of each GD and Al monolayer using the CM for comparison.

**Figure 4 bioengineering-11-00366-f004:**
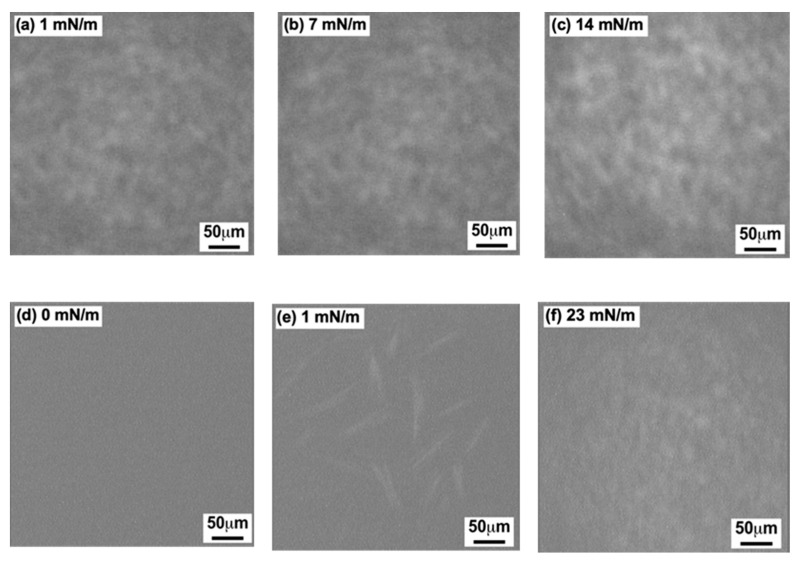
BAM images of GD and Al monolayers on the water surface at 26 °C using the DM. (**a**–**c**): GD (dimer) monolayer; (**d**–**f**): Al monolayer. Inserted surface pressures correspond to the value in each *π-A* isotherm curve shown in Figure 2.

**Figure 5 bioengineering-11-00366-f005:**
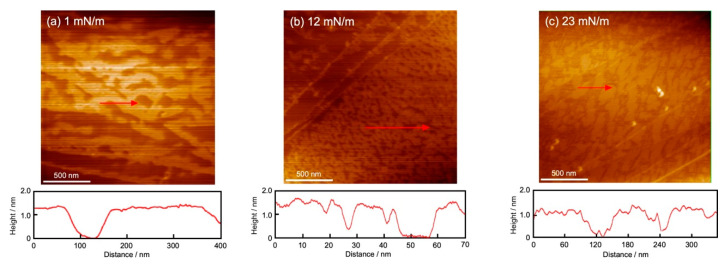
AFM images of GD and Al monolayers scooped on the HOPG surface after each monolayer formation at 26 °C using the DM. (**a**,**b**): GD (dimer) monolayer; (**c**): Al monolayer. Observation range: 2 µm × 2 µm. Inserted surface pressures correspond to the values in each π*-A* isotherm curve in Figure 2. Red arrow and cross-section profile: thickness (height) of each GD and Al monolayer in the arrow range.

## Data Availability

Data is contained within the article.

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
