# Peer review of "Morphology Observation of Two-Dimensional Monolayers of Model Proteins on Water Surface as Revealed by Dropping Method"

_bioengineering, 2024, doi:10.3390/bioengineering11040366_

Round 1
Reviewer 1 Report
Comments and Suggestions for Authors
Authors use a monolayer model to investigate the properties of gramicidin and alamethicin proteins monolayers formed on water through morphological, surface tension, Brewster angle microscopy and atomic force microscopy analyses. They conclude that that the dropping method is a useful technique that can reveal the two-dimensional aggregation properties of the model biological membrane components on the water surface.
I consider that the topic addressed by the authors is not relevant. Analyzing whether there are changes in the orientation of the proteins (gramicidin-D and alamethicin) in a monolayer formed on water has no relevant purpose.
Although authors mention in the introduction section that the interactions of biological membranes with proteins, peptides or other substances affect their functions as a barrier, substance transport, molecular recognition, and signal transduction they do not provide a theoretical framework that highlights the importance of this analysis or that indicates the practical application of its findings.
I assume that it provides little to the study area considering that they use 2 proteins with different characteristics and properties (gramicidin-D, double helix, and alamethicin, single helix) to see if there are differences in the morphology of these protein monolayers.
Unless the authors propose each of these proteins as representative of other molecules, I would consider that the impact of the work would be greater.
The authors do not mention or describe that they have carried out the indicated measurements using only the solvents of the 2 proteins (methanol and methanol/chloroform).
Conclusions are consistent with the results.
The cited twenty-nine references are appropriated.
There are no tables of results in the manuscript and regarding the 4 figures that show the results of the analyzes carried out, I only consider that there is a lot of space between the images in figure 4.
Observations.
Please check wording. Some examples of bad writing are as follows:
“… and is known to act as an ion channels by assembling 3-12 molecules…”
“The structure of GD and Al was shown in Figure 1.”
“…using various physicochemical method.”
In the text corresponding to lines 147 to 153 seems that abbreviations are missing (blank spaces). A similar situation is observed in other paragraphs, which makes it difficult to understand the text.
It is hard to understand the results section because of the blank spaces in this section.
There are a strikethrough letters or words in lines 175, 206, 228, 233, 285, 293, 338.
A bold letter is in line 210.
The images in figure 4 are very far apart from each other, they could be closer.
Comments on the Quality of English Language
Please check wording. Some examples of bad writing are as follows:
“… and is known to act as an ion channels by assembling 3-12 molecules…”
“The structure of GD and Al was shown in Figure 1.”
“…using various physicochemical method.”
Author Response
All the corrections have been removed. Please find PDF file of the response to the comments and queries.

Reviewer 2 Report
Comments and Suggestions for Authors
The work “Morphology Observation of Two-dimensional Monolayers of 2 Model-Protein on Water Surface as Revealed by Dropping Method” is an interesting study. However, it is written in a difficult to understand language. There are missing letters (likely, symbols), e.g. in lines 147, 150-152. Some data are missing, too. In the Abstract section it is stated: “CD spectra of GD/methanol solution showed GD molecules existed in dimer form”. However, no any CD spectra is presented. Taking all the above I suggest:
1. Please, provide CD spectra in the body of the paper and say clearly, what is the structure of GD dimer, is it head-to-head, or double helix and, please, provide references to argument either choice;
2. Improve English;
3. Fill in all missing symbols in the text;
4. Remove too general sentences in the Introduction, e.g. about 20 types of amino acids. Is alamecithin composed solely of standard amino acids?
Comments on the Quality of English Language...perform their functions using the surrounding water of them as a medium...
Author Response
All the corrections have been removed (including grammar) in the revised version. Please find enclosed PDF of the response to the comments and queries.

Round 2
Reviewer 1 Report
Comments and Suggestions for Authors
The authors addressed the observations of this reviewer
Author Response
I have addressed all the issues raised by Reviewer 1 in round one already. In round 2, there are no comments to attend.
Reviewer 2 Report
Comments and Suggestions for Authors
See attached file.

Comments on the Quality of English LanguageSee attached file.
Author Response
Dear Reviewer,
I have addressed all the issues mentioned. Please see the enclosed PDF. I hope the paper is good to publish.
Thank you and regards,
Dr. Zameer Shervani, Ph.D.
